# Exploring Cultural Hybridity Branded by Convergence and Syncretism in the Characteristic Features of the Pentecostal Charismatic Churches in Zimbabwe: Implications for Spiritual and Material Well-Being

Francis Marimbe 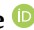

Development Studies, School of Built Environment and Development Studies, College of Humanities, University of KwaZulu-Natal, Durban 4000, South Africa; francis@drisa.net

**Abstract:** When applied to Pentecostalism in Zimbabwe, the concept of cultural hybridity provides a framework for understanding how global religious movements can adapt to and incorporate local cultural elements. This process results in a unique form of religious expression characterised by convergence and syncretism, reflecting cultural and religious identity's dynamic and fluid nature. This hybridity in religious practice is a testament to the ongoing, interactive cultural exchange and adaptation process. This article delves into the intricate cultural hybridity, convergence, glocalisation and syncretic tendencies within the characteristic features of New Religious Movements (NRMs) in Harare, Zimbabwe, illuminating their multifaceted role in addressing spiritual and material needs. Through a comprehensive exploration of selected NRMs that emerged from the Catholic Church in Zimbabwe, including Prophetic Healing and Deliverance Ministries and Grace Oasis Ministries, this article unravels the central role of prophets and pastors in shaping the fundamental ethos of these religious entities. A striking and thought-provoking parallel emerges between the hallmark features of these NRMs and the tenets of African Traditional Religion and many other religious traditions. This parallel extends to practices such as exorcism, worship, healing, and deliverance, thus manifesting a profound form of religious expression informed by cultural hybridity, convergence, syncretism, and glocalisation. While there are ambiguities around scholarly debates on the definition of these terms, the article delves deep into the intricate religious elements embedded within the NRMs' characteristic features, such as hymns, modes of worship, healing rituals, and deliverance ceremonies. These elements are tangible manifestations of their unique position at the crossroads of diverse belief systems. The cultural hybridity, convergence, syncretism, and glocalisation tendencies within NRMs offer gateways to invaluable networks, fostering social cohesion and the sharing of critical information. Consequently, these characteristics have become instrumental in the holistic development of individuals and communities within the vibrant religious landscape of Harare. Thus, this article provides profound insights into the nuanced dynamics of NRMs in Zimbabwe, shedding light on their various dimensions. It contributes substantially to our comprehension of the intricate interplay between spirituality, material prosperity, and the rich tapestry of religious traditions in Harare and the broader context of religious studies.

**Keywords:** Pentecostalism; cultural hybridity; convergence; syncretism; glocalisation; new religious movements; African Traditional Religion; Zimbabwe

## 1. Introduction

In the ever-evolving landscape of global religious practices, the phenomenon of Pentecostalism, characterised by its emotive and expressive form of Christianity, intertwines interestingly with the concepts of cultural hybridity, convergence, glocalisation, and syncretism. This blend is particularly notable in Harare, Zimbabwe. Pentecostal–Charismatic

churches, hereby referred to as New Religious Movements (though not all Pentecostal–Charismatic churches are NRMs), rooted in the Christian tradition, exhibit unique features which reflect a significant intersection between spirituality and material well-being, which is crucial for understanding the broader implications of religious movements in contemporary society. Considering that there is an overlap and no clear typology of NRMs in Zimbabwe, the term NRMs in this article will refer to Prophetic Healing and Deliverance (PHD) Ministries and Grace Oasis Ministries (GOM), which have emerged from the Catholic Church in Zimbabwe.

This article presents three main arguments. The first argument highlights that the distinct features of the selected New Religious Movements (NRMs) exhibit cultural hybridity, convergence, glocalisation, and syncretism. These features significantly contribute to the fulfilment of spiritual and material needs, with prophets or pastors playing a pivotal role in defining the core ethos of these religious groups. These NRMs represent a fusion of Christian elements with aspects of African Traditional Religion (ATR), particularly in practices such as exorcism, worship, healing, and deliverance. The second argument draws a parallel between the characteristics of the chosen NRMs and the tenets of ATR and other religious beliefs. This parallelism is seen in practices that demonstrate religious expression shaped by cultural hybridity, convergence, glocalisation, and syncretism. Moreover, the religious aspects within the NRMs' characteristics are concrete examples of their unique positioning at the intersection of various belief systems. The third argument posits that the cultural hybridity, convergence, glocalisation, and syncretism in the NRMs provide pathways to invaluable networks. These networks play a crucial role in enhancing social cohesion and facilitating the exchange of essential information, contributing to the comprehensive development of individuals and communities within the religious milieu of Harare, Zimbabwe.

Therefore, this article illustrates a complex interplay between spirituality, material prosperity, and the rich diversity of religious traditions in Harare, which holds broader implications for religious studies. By examining these NRMs, the article aims to shed light on the controversial and divergent hypotheses surrounding cultural hybridity, convergence, glocalisation, and religious syncretism, particularly in the context of NRMs. The findings reveal that NRMs' richness rests in their ability to take root and translate their ideas in diverse environments, contributing to community development and individual empowerment and reshaping Harare's religious and socio-economic landscape.

## 2. History of Pentecostalism and NRMs in Zimbabwe

Pentecostalism, which has roots in America, significantly evolved and adapted as it spread to Africa, particularly Zimbabwe, intertwining with various waves of Christianity and reflecting a unique blend of religious elements. This transformation led to a distinctive form of faith practice marked by phases of development, each characterised by different focuses and teachings. From its origins in America, Pentecostalism was shaped initially by missionary evangelism and classical Pentecostalism (Meyer 2007). This laid the foundation for what later became known as the 'Third Wave' Pentecostalism, noted for its emphasis on the 'gospel of prosperity'. The movement gained momentum during the healing revivals of the 1950s in the USA, led by influential evangelists such as Essek William Kenyon, Kenneth and Gloria Copeland, and Oral Roberts (Cornelio and Medina 2020). These figures championed the prosperity gospel, advocating that divine favour is manifested through wealth and health. This gospel, intertwined with the New Religious Right Movement in the USA during the 1970s, shifted in its socio-political agenda as it spread to Africa.

Bowler (2013) and Swoboda (2015) noted that the prosperity gospel comprises faith, wealth, health, and victory. It teaches that Christ's suffering and death have met all human needs, allowing believers to share in his victory (Cornelio and Medina 2020). However, critics argue that this doctrine diverts focus from African structural issues to individual faith and material blessings (Gifford 1991). Contemporary Pentecostalism in Africa focuses on African Pentecostalism, 'Third Wave' Pentecostalism or Neo-Pentecostal churches or New

Religious Movements, which represent a unique fusion of Christianity and African cultural elements, characterised by modernity, relaxed norms, internationalism, and innovative worship methods (Togarasei 2011; Gifford 2015; Asamoah-Gyadu 2015). This form of Christianity is often centred around charismatic leaders who embody the church's vision.

In Zimbabwe, Pentecostalism was initially facilitated by the Apostolic Faith Mission, forming various Pentecostal churches, each highlighting the gospel of prosperity (Chitando 2013; Matikiti 2017; Mapuranga 2013; Mumford 2012). Post-independence, the socio-economic situation fostered the growth of Neo-Pentecostal churches or NRMs inspired by financial contributions to prophets or pastors who claim to be God's representatives. Influenced by American counterparts, Neo-Pentecostalism particularly resonated with the youth through its modern worship styles. The transformation of these movements from small fellowships to large denominations involved leadership changes and increased emphasis on status symbols.

The 1980s were pivotal for the growth of NRMs in Zimbabwe, with many churches emerging under the leadership of Zimbabwean and Nigerian founders. These leaders often trained under prominent West African preachers known for performing miracles, such as T.B Joshua and Pastor Chris Oyakhilome (Vengeyi 2013). This period saw a blend of miraculous healing and prosperity gospel, resonating with the African socio-economic and political contexts. Post-2008, a new dimension of Pentecostalism emerged in Zimbabwe, characterised by the rise of young Pentecostal prophets (Gukurume 2021). This development is linked to the 'spirit type' prophets of Apostolic, Zionist, and charismatic African Independent Churches (AICs) from the 1930s. Gunda (2012) and Zimunya and Gwara (2013) argue that the new wave of prophets promoted a gospel of prosperity coupled with miracles, appealing to a society grappling with hyperinflation, economic crisis, and hopelessness.

The 2008 period also witnessed the rise of prophets such as Emmanuel Makandiwa (United Family International Church), Uebert Angel (Spirit Embassy), Passion Java of Kingdom Embassy, Adventure Mutepfa of Revival Centre World Ministry and Oliver Chipunza of Apostolic Flame Ministries of Zimbabwe (Vengeyi 2013, p. 28). Within this article, the focus will be on Prophetic Healing and Deliverance Ministries of Walter Magaya and Grace Oasis Ministries of Pastor Tinashe. These new churches arose during 2012 and 2015, respectively, a period often characterised as the lowest point in the Zimbabwean crisis (Chitando et al. 2015, p. 2). To date, Zimbabweans are battling with such economic crises, worsened by the political crises. However, Pentecostalism's journey from America to Zimbabwe represents an evolution from evangelical roots to a more syncretic, prosperity-oriented movement within the African context. This evolution mirrors the adaptability of Pentecostalism to diverse cultural and socio-economic environments. As a central tenet, the prosperity gospel underscores the intricate relationship between faith, material success, and individual effort in shaping the religious landscape of contemporary Africa. In Zimbabwe, the rise of NRMs reflects both a response to and a reflection of the nation's socio-cultural dynamics, offering solace and hope amidst economic and political challenges.

## 3. African Traditional Religion (ATR)

African Traditional Religion (ATR) is a diverse and complex spiritual system deeply ingrained in the cultural fabric of various African societies (Humbe 2020; Falola 2022). It is not a unified religion but a broad spectrum of indigenous beliefs and practices that vary significantly across African ethnic groups and regions (Van Rooyen 2019). Its main features include polytheism and Ancestor worship. It is believed that ATR involves belief in multiple deities, often associated with natural elements and phenomena. Ancestor worship is also central, with ancestors being revered and believed to influence the living world (Ushe 2022; Ephirim-Donkor 2021). It is also pertinent to note that ATR is transmitted mainly through oral tradition, encompassing myths, proverbs, and folktales that convey moral and ethical teachings, cosmology, and history. There are also rituals, including offerings, sacrifices, and festivals, which are vital in ATR for maintaining harmony between

the spiritual and physical realms (Alifa 2023). These ceremonies often mark significant life events, and seasonal cycles witnessed in communities, with a strong emphasis on ATR. ATR teaches that practices and beliefs are intertwined with communal life, highlighting the interconnectedness of individuals within their community and with the natural world (Humbe 2020). Shamanism and traditional healing are crucial in ATR, mediating between the spiritual and physical worlds. Shamans and traditional healers are often responsible for healing, spiritual guidance, and maintaining the community's well-being (Jakobsen 2020; Mullinder 2023).

Interestingly, ATR promotes living in harmony with nature, often attributing sacred qualities to natural sites and elements, reflecting a deep respect for the environment. It also provides a framework of moral and ethical values, emphasising virtues such as respect, community responsibility, and living in harmony with others and the environment. What is clear is that ATR's adaptability and resilience are evident in its ability to coexist and syncretize with other religious systems, reflecting the dynamic and evolving nature of African spiritual practices.

## 4. Cultural Hybridity, Convergence, Syncretism and Glocalisation

Scholarly debates on syncretism, hybridity, and convergence within African Pentecostalism are complex and multifaceted. Each term encapsulates different aspects of how religious practices and beliefs interact with local cultures. The complexity and ambiguity around the debates of the definitions of these terms also present challenges related to understanding the religious practices and how they constitute syncretic or hybridised practices among Pentecostal–Charismatic Churches. Syncretism is a specific type of cultural blending. It traditionally refers to combining different religious beliefs or practices (Pandian 2006; Greenfield and Droogers 2001). However, syncretism can involve fusing diverse cultural elements in a broader cultural context. This fusion results in a new form that bears traces of its original components but is distinct. Ogbona and Agaba (2021) argue that syncretism in religious studies refers to blending different religious beliefs and practices. Within African Pentecostalism, this often involves incorporating elements of traditional African religions into Christian practices.

Regarding cultural hybridity (discussed below), syncretism can be seen as a more profound, integrative blending process. It is also about creating something new that cannot be easily disentangled into its original parts. For example, many modern cultures practice religious or cultural rituals that have evolved over centuries, incorporating elements from various sources to create practices unique to their current cultural context. A considerable debate among scholars about the implications of syncretism shows that syncretism represents a loss of the 'purity' of religious traditions, potentially diluting Christian theology with non-Christian elements (Clarke 2022; Gbode 2021; Ndhlovu 2020). Other scholars view it as a natural and inevitable cultural and religious interaction process which can enrich and deepen the religious experience. Within African Pentecostalism, syncretism is often observed in incorporating beliefs in spirits, ancestors, and traditional healing practices into the Christian faith. Scholars debate whether this represents a pragmatic adaptation of Christianity to local contexts or a compromise of Christian doctrinal purity. Hence, there is ambiguity in the scholarly definition of syncretism, which also presents challenges when it comes to understanding if the religious practices (to be discussed in Section 7) constitutes syncretic tendencies within NRMs.

In cultural studies, the concept of hybridity refers to creating new cultural forms by combining elements from different cultures. In religious contexts, this can mean blending religious practices with local cultural norms (Ackermann 2012). Ackermann (2012) argues that cultural hybridity refers to blending elements from different cultures. This blending can occur in various forms—through art, literature, language, customs, and other cultural practices. It is a process that often happens when different cultures come into contact with each other, whether through migration, colonisation, globalisation, or digital connectivity. In Ackermann's context, cultural hybridity might be viewed as a dynamic and ongoing

process, not just a simple mixing of two distinct cultures to create a third, 'hybrid' culture, but a more complex interaction in which cultural elements are continuously exchanged, transformed, and renegotiated. This concept challenges traditional notions of culture as fixed or pure, instead highlighting the fluid and evolving nature of cultural identities (Bhandari 2021). However, the debate around hybridity often centres on its implications for cultural identity and authenticity. Some scholars see hybridity as a positive force, arguing that it leads to greater cultural diversity and adaptability. Others worry it may erode traditional cultures and identities (Lee 2022; Mohiuddin 2023). Concerning African Pentecostalism, cultural hybridity can be seen in how both global Pentecostalism and local African cultural contexts influence worship styles, theological interpretations, and church governance. This has led to unique forms of Pentecostalism that are distinctly African, though the complexities around cultural hybridity play a role in our understanding of hybridised practices among NRMs.

Convergence in the context of cultural hybridity refers to combining different cultural and religious elements to form a cohesive whole (Pieterse 2019). It is often associated with the processes of globalisation and transnational movements and linked to the effects of globalisation and technological advances, which have accelerated the process of cultural exchange and interaction. It focuses on how disparate cultural practices, ideas, or symbols can merge to create new forms of expression or cultural norms (Cleveland et al. 2022). In cultural studies, convergence might be observed in global media, in which different cultural products (films, music, television shows) incorporate elements from various cultures, making them more universally appealing or accessible. Scholars debate the extent to which convergence in religious practices represents a form of cultural homogenisation versus a form of cultural diversification. Some argue that convergence leads to a loss of local cultural specificities, while others see it as a process that enriches religious practices by offering diverse perspectives (Holton 2000; Cleveland et al. 2022). This clearly shows that the definition of convergence needs to be more precise. In African Pentecostalism, convergence is evident in how global Pentecostal practices are adapted to local contexts. This includes the adoption of global Pentecostal doctrines while maintaining distinctive African religious and cultural elements.

Ackermann's discussion of cultural hybridity and the concepts of convergence and syncretism highlights cultures' dynamic and ever-changing nature. These concepts challenge static views of cultural identity, showing how cultural exchange and interaction are central to the development of societies. They emphasise that cultures are not isolated or unchanging but are continually influenced and reshaped by their interactions with other cultures, leading to new forms of cultural expression and identity. There is a central theme in the debate on how these processes (syncretism, hybridity and convergence) impact cultural identity. African Pentecostalism is a clear example of how global religious movements interact with local cultures, leading to new forms of religious identity that are neither entirely African traditional nor entirely Western Pentecostal. In addition to the theme of cultural identity, there is theological purity versus cultural relevance. There is a tension between maintaining theological purity and adapting to cultural relevance. This tension is particularly pronounced in discussions around syncretism. However, the debate around syncretism, hybridity, and convergence within African Pentecostalism highlights the complex ways global religious movements interact with local cultures. These interactions result in unique forms of religious expression that challenge traditional notions of religious and cultural purity, raising essential questions about identity, authenticity, and the nature of religious practice in a globalised world.

Joel Robbins (1998, 2004, 2017), an anthropologist known for his work on cultural change, particularly in the context of Christianity among indigenous populations, provided a relevant perspective on the concept of 'glocalisation' in religious traditions. 'Glocalisation' is a term that refers to the way global phenomena are reinterpreted and adapted in local contexts, leading to unique, localised expressions of globally recognised phenomena, including religious practices (Robbins 2017; Roudometof 2016). Robbins' work, especially

his studies on the Urapmin community in Papua New Guinea and their conversion to Charismatic Christianity, offers valuable insights into this process. Robbins argued that while there is a significant shift in religious practices, certain aspects of cultural identity and traditional beliefs continue to influence the Urapmin community's interpretation and practice of Christianity. In addition, he noted the notions of syncretism and hybridisation, touching upon how indigenous communities blend elements of their traditional religious practices with Christianity. This syncretism is not just a simple combination but a complex and dynamic process, leading to a unique form of Christianity that is deeply influenced by the local cultural milieu. More so, a significant focus of Robbins' research is on conversion, particularly how the Urapmin people understand and negotiate their Christian identity while retaining aspects of their indigenous identity. He delves into how conversion impacts social structures, moral systems, and individual identities within the community. Joel Robbins' anthropological work provides a nuanced understanding of glocalisation within religious traditions. His studies demonstrate that transforming religious practices in local contexts is not a one-way process of global imposition but a complex interplay of global and local influences, in which local communities actively shape their religious and cultural landscapes. This perspective challenges simplistic notions of cultural imperialism and highlights the dynamic nature of cultural and religious change.

When applied to Pentecostalism in Zimbabwe, Ackermann's concept of cultural hybridity provides a framework for understanding how global religious movements can adapt to and incorporate local cultural elements. This process results in a unique form of religious expression characterised by convergence and syncretism, reflecting cultural and religious identity's dynamic and fluid nature. This hybridity in religious practice is a testament to the ongoing, interactive cultural exchange and adaptation process.

## 5. Materials and Methods

The study employed a qualitative research methodology to explore cultural hybridity, convergence, glocalisation, and syncretic aspects of NRMs in Harare, Zimbabwe, from January 2020 to December 2022. Within the qualitative methodology, the study used an ethnographic approach to collect data through interviews, focus-group discussions (FGDs) and observations. More so, data collection was rooted in a snowball sampling approach initiated by a leader from one of the selected churches. The leader facilitated introductions to other congregants, thereby expanding the research network. The respondent pool included members from Prophetic Healing and Deliverance (PHD) Ministries, with an estimated congregation size between 900,000 and 1 million, and Grace Oasis Ministries (GOM), comprising about 500 followers. Both ministries consented to use their real names in the study, while pseudonyms were employed for individual research participants to ensure confidentiality.

The primary data collection process involved in-depth interviews with eighteen respondents, encompassing four ministers, four church elders, three gatekeepers, three staff members, and four ministry members. Over three years, the researcher conducted eight non-participant observations and three focus-group discussions to understand the subject matter comprehensively. Additionally, four research campaigns were executed, wherein baseline data were collected from selected interviewees through structured interviews. These interviews were audio-recorded and later transcribed for analysis.

The collected data were analysed using thematic analysis, informed by the researcher's extensive background in studying NRMs in Zimbabwe. This long-term understanding of the selected NRMs significantly contributed to the depth and richness of the analysis. Moreover, the findings were further enriched by a thorough review of secondary literature, including published articles, books, and information from the official websites of the involved churches. The focus and scope of the research were explicitly tailored to the selected NRMs in Zimbabwe, offering a detailed and contextual understanding of their practices within the framework of Pentecostalism. This methodological approach allows

for the replication and extension of the study by other researchers interested in the dynamic interplay between religious traditions and modern faith practices in Zimbabwe and beyond.

## 6. Unique Features of the NRMs in Zimbabwe

The characteristic features of the NRMs in Zimbabwe are similar to those of the African context. The young and famous prophets, who pay more attention to miraculous wealth and healing, have enticed many poor, young, urban middle-class, elite, and educated Zimbabweans who experience severe economic challenges (Togarasei 2011, p. 341; Mangezi and Manzanga 2016). Prophets claim to perform 'extraordinary' miracles which contradict nature. Between 2010 and 2013, Zimbabwe witnessed a range of 'miracle money' in which money was believed to fill the pockets, wallets, and hands of believers with gold nuggets. Prophets also claim to heal cancer, HIV, and AIDS, and raise the dead. These prophets have been known for performing other miracles such as instant weight loss, complete regrowth of teeth that have fallen out, miracle babies, refuelling cars without going to the service station, and making predictions of events that will happen in the near or distant future. Such miracles are closely linked to the one claimed by TB Joshua of Synagogue Church of All Nations, pastor Victor Kusi Boateng of Ghana, and Pastor Chris Oyakhilome of Christ Embassy (Vengeyi 2013, p. 30). Healing and deliverance also feature prominently in the NRMs' worship and naming of some churches. Prophetic Healing and Deliverance Ministries and Grace Oasis Ministries are of interest in this study.

Another unique feature of NRMs in Zimbabwe is what Cornelio and Medina (2020, p. 65) call the prosperity ethic, popularly known as the gospel of prosperity. This new prosperity ethic is backed by a religious belief in individual work ethic that promises financial returns. It has two features, which involve sacralising self-help and celebrating consumption. Furthermore, it is individualistic as opposed to a collective effort. It sacralises work ethic with the conviction that God wants to bless his people so they can be wealthy and successful while ensuring financial growth and freedom from debt (Cornelio and Medina 2020). What is unique about this phenomenon is that the old prosperity gospel is miracle-oriented, providing a message of hope for the poor and preaching breakthroughs by giving a positive confession while asking believers to simply believe, profess, and give money to the prophets or pastors, thereby witnessing miraculous blessings (Mangena and Mhizha 2014, p. 138). Such an old prosperity gospel is still common within the Zimbabwean NRMs. However, Cornelio and Medina (2020, p. 70) see the new prosperity ethic as teaching believers to adopt practical skills related to investment, financial, management and religious innovations that address economic insecurities and build on personal aspirations for spiritual growth and material success.

The prosperity ethic is an unapologetic enjoyment of the good life here and now while responding to the situation of an emerging middle class with emphasis on self-help and consumption to benefit from the economy using biblical and Christian principles (Cornelio and Medina 2020, p. 71). The above-proposed prosperity ethic, which seems to be unique within the NRMs in Zimbabwe, and the well-known prosperity gospel seem to share the difference in content. The old or familiar prosperity gospel seems to rely heavily on the promise of a financial miracle, which is activated through the power of confession and giving, and the prosperity ethic emphasises financial growth through self-help and other practical tips about investment and resource management. As a result of these emphases, the old prosperity gospel and the new prosperity ethic have attracted different audiences. The former is a message of hope for the poor. The latter works for the aspirational middle class (Togarasei 2011, p. 341; Mangezi and Manzanga 2016; Mangena and Mhizha 2014), though I believe this phenomenon targets the poor and middle class in Zimbabwe.

## 7. Characteristic Features of NRMs and Tenets of ATR

Gukurume (2021, p. 28) argues that NRMs attend to spiritual and material needs. In Pentecostal churches in Zimbabwe, cultural hybridity, convergence, syncretism, and glocalisation are reflected in how the selected NRMs adapt and integrate local cultural

elements with Christian practices. In addition, these concepts are vividly manifested in various aspects of church life and practice, such as worship style, theological interpretation, alleged miracles, use of media, announcements and testimonies, and churchpreneurs, sowing the seed and community engagement as demonstrated below. This section covers the article's first argument that the selected NRMs' characteristic features exhibit cultural hybridity, convergence, glocalisation, and syncretic tendencies. It highlights how the selected NRMs' characteristic features are critical in their operations and socio-economic aspects. It also covers the second argument, which states that there is a parallel between the characteristics of the selected NRMs and the tenets of ATR and other religious beliefs.

*7.1. Hymns*

The praise and worship activities that percolate around hymns and their meanings are integral to the NRMs. There are cultural hybridity-blending elements and glocalisation between the features of NRMs and ATR in the context of hymns and worship practices that are fascinating and multifaceted. These elements reflect a dynamic interplay between traditional African spirituality and contemporary religious expressions. Both PHD and GOM tend to have similar hymns, which can be explained by the fact that the latter was born from the former. These hymns include '*Makomborero hobho, tambira Jehovah*', which means that blessings are abundant and that one has to dance for the Lord. At GOM, songs are sung in Shona, Ndebele, and English. An imitation of the Nigerian accent in some songs has been noted, thereby demonstrating reinterpretation and adaptation of global phenomena into a local context, leading to a unique localised expression of music as noted by Robbins (2017) and the famous song, 'Jesus, You Love Me Too Much', which was sung at both GOM and the PHD Ministries. Such glocalisation elements continue to influence some NRMs in Zimbabwe. The use of multiple languages (Shona, Ndebele, and English) and even the imitation of Nigerian accents in songs highlight the adaptability and inclusivity of NRMs and the influence of cultural identity and traditional beliefs. This linguistic diversity mirrors ATR's flexibility and capacity to encompass various cultural elements within its practices. Such a characteristic feature of NRMs and its inclusivity played a role in addressing the spiritual needs of PHD Ministries and GOM followers.

Kalu (2008, p. 15) argues that hymns in NRMs are so electric in their doctrinal emphases that their music sounds as if it is being performed at a disco or club. The energetic and entertaining nature of NRMs' worship services atmosphere resonates with ATR's emphasis on vibrant spirituality. This engaging style of worship caters to the emotional and spiritual needs of the congregation, similar to how ATR practices engage and involve community members. Ackermann (2012) argues that hybridity influences worship styles, theological interpretation, and church governance. The worship style of the selected NRMs showed a blending of religious practices and cultural norms being continuously exchanged, transformed, and renegotiated. Following an analysis of the complete services of worship at the PHD Ministries and GOM, it was noted that the believers sang praise and worship songs that were modern and entertaining, some similar to the Democratic Republic of the Congo (DRC) type of dance called 'ndombolo dance', showcasing the merging of cultural practices which created a new form of religious expression, as noted by Cleveland et al. (2022). Togarasei (2010, p. 356) has also noted that the DRC's famous *ndombolo* or *kwasa-kwasa* dance is incorporated into worship. It was pertinent to note that the NRMs' use of hymns and music that blend traditional African rhythms with modern styles is a primary example of cultural hybridity, syncretism, glocalisation, and convergence.

The incorporation of 'ndombolo' and 'kwasa-kwasa' dances from the DRC into worship services exemplifies how African cultural elements are seamlessly integrated into contemporary religious practices. This mix of traditional dance with modern, secular styles resonates with the youthful congregation, reflecting a harmonious blending of the old and the new or different religious practices (Ogbona and Agaba 2021). More so, the NRMs' ability to 'read the signs of the time' and incorporate secular-youth-type dances into their ministries reflects ATR's adaptability and responsiveness to its adherents' changing needs

and contexts, as noted by Robbins (2017). This aspect highlights religious practices' ongoing evolution and relevance in addressing the contemporary spiritual landscape.

Within the above context, it was noted that the NRMs read the signs of the time and offered the secular-youth type of dance they adopted into their ministries. Horton (1971, p. 86), writing about African conversion, made a related point that songs and prayers within the NRMs 'correctly specify and take account of the various visible and invisible forces at work in any given situation'. This is true based on Sharpe's (2003, p. 110) concept of 'religious syncretism'. The worship of the selected NRMs coincided with traditional African cosmology. In other words, the NRMs' worship practices align with African traditional cosmology, suggesting that they are not merely adopting ATR practices superficially but integrating ATR's underlying worldview and spiritual understanding into their worship. The NRMs' worship has always been a vibrant type of spirituality that stimulated the faith of the believers, which surprised the researcher as an observer. What is clear, however, is that the worship practices of NRMs in the context of hymns and music exhibit a rich tapestry of syncretic and convergence elements, blending ATR's spirituality, communal involvement, and adaptability with contemporary religious expressions. This cultural hybridity preserves traditional African elements and revitalises, transforms and renegotiates them (Ackermann 2012), making them relevant and appealing to modern congregations. Cultural hybridity within the selected NRMs blends traditional African music, dance, and attire with Christian worship styles. This has created a unique worship experience resonating with the local populace addressing their spiritual needs. More so, NRMs worship practices demonstrate a fusion of different expressions, creating a distinctive form of worship that is neither purely African traditional nor entirely Western Christian.

*7.2. Prayer and Raising of Hands*

Raising hands during prayer and singing has been noted as one of the characteristic features of the NRMs, which also suggests an interplay of global and local influences and the creation of new cultural forms between NRMs and ATR in the embodied and communal nature of worship, as well as a focus on addressing everyday challenges through spiritual means. The singer in the selected NRMs encouraged people to raise their hands at the end of each song, and a 'thank you, Jesus' was always expressed. Pastors would shout, 'Clap hands for Jesus' (Hollenweger 1992, pp. 7–17). Loudspeakers and cameras were prominently featured during preaching, praise, and worship. NRMs' prayer activities are emotional, enthusiastic, and entertaining, with every member actively participating in the liturgy (Mukwakwami 2010, p. 11; Anderson 2001, p. 171; Hollenweger 2004, pp. 125–37). What is evident is that the enthusiastic and inclusive nature of prayer activities in NRMs, in which every member participates actively, resonates with ATR's communal approach to worship. In ATR, community involvement is crucial, and spiritual practices are often a collective activity (Humbe 2020). Similarly, NRMs emphasise communal engagement and collective spiritual experiences during prayer (Herzog et al. 2020). More so, incorporating loudspeakers and cameras in NRMs' services represents a modern adaptation of traditional practices in the local context, leading to a unique localised expression as noted by Robbins (2017). This use of technology to enhance worship reveals how NRMs, like ATR, evolve and adapt to contemporary contexts while maintaining core spiritual practices.

It was observed that the worshippers within the selected NRMs recited prayers as if they were reciting poems, and some believers would jump up and down, throwing themselves down and shouting for God's help. The fervent and passionate nature of prayers witnessed within the selected NRMs involving shouting and intense expressions shares similarities with ATR's emphasis on conviction and intensity in spiritual practices. This combination of different religious elements formed a cohesive whole, strengthening the connection with the divine and ensuring prayers were heard, a concept prevalent in ATR and NRMs. In addition, the practice of raising hands, clapping, jumping, and shouting during prayers and worship in NRMs reflects hybridity and glocalisation with ATR's embodied and emotional forms of worship (Adewole 2023; Buertey 2023).

In many ATR practices, physical expressions such as dancing, clapping, and bodily movements are integral to connecting with the spiritual realm (Mtshali 2020). This physicality in worship, embraced by NRMs, aligns with ATR's emphasis on expressing spirituality through the body. However, is is difficult to determine if these practices constitute syncretic or hybridised practices among NRMs due to an ambiguity around the scholarly debates on the definition of hybridity, convergence, glocalisation, and syncretism. More so, the prayers are based on convincing God to intervene in specific everyday problems, including social and economic challenges. The focus of NRMs' prayers on everyday social and economic challenges mirrors ATR's practical approach to spirituality. In ATR, spiritual practices are often directed towards seeking guidance, protection, and intervention in daily life issues. Campbell (2020) argues that churches continue this tradition by encouraging prayers that directly address their congregants' immediate needs and concerns. Prayer and shouting seem to be encouraged to ensure God hears one's requests.

### 7.3. Alleged Miracles

The portrayal of alleged miracles in NRMs and their connection to ATR reflects the blending of religious practices with local cultural norms (Ackermann 2012). It also reflects a combination of different cultural and religious elements to form a cohesive whole (Pieterse 2019) that emphasises spiritual healing and practical solutions to life's challenges. It was observed that in the back of the PHD Ministries offices, there was a place where so many wheelchairs, crutches, and walking sticks that belonged to those who were allegedly healed were displayed as testimonies of healing, which mirrors ATR's focus on physical and spiritual healing. The display of such equipment mirrors the merging of cultural symbols and practices, creating a new form of expression. In ATR, healing rituals are common; traditional healers use a combination of spiritual and herbal remedies to treat ailments (Mothibe and Sibanda 2019; Ozioma and Nwamaka Chinwe 2019). Similarly, Kgatle and Thinane (2023) believe that new prophetic churches emphasise miraculous healings, aligning with the traditional belief in supernatural interventions for health problems. Such cultural and religious exchange and interaction elements show the adaptation of NRMs practices to the local context.

Stories of people who suffered from a back problem, those who had their leg amputated, and several other ailments were ostentatiously displayed on the walls. They highlighted people's lives before and after the alleged deliverance. This type of deliverance suggested being rescued, liberated, or set free from captivity, evil, or danger through healing and deliverance, which promoted quality health and peace of mind. This was similar to what Asamoah-Gyadu (2007, p. 398) observed in NRMs in Ghana. Considering the above, testimonies and stories about people's lives before and after deliverance in NRMs are reminiscent of ATR's oral tradition. Day (2021) argues that storytelling is vital for preserving history, teachings, and testimonies of spiritual experiences. Similar to ATR, NRMs adapt this practice to showcase stories of healing and transformation, providing a modern context to this traditional practice. This demonstrated the understanding of cultural hybridity, which brings different cultures together, continuously exchanging, transforming, and renegotiating to ensure value and meaning within NRMs.

Upon visiting the PHD Ministries carrying out interviews and observations, no single incident of the physical healing of those walking with crutches or walking sticks was witnessed. Stories of healing are used as advertisements to lure new followers. One of the believers, highlighting how they joined the PHD Ministries, argued, 'I joined the PHD Ministries after I heard of the miracles which the prophet performs. On the first day I joined, I felt at home where other believers linked me with those close to my house.' Using healing stories as advertisements in NRMs is a modern adaptation of ATR's practice of building credibility through testimonies of effective spiritual interventions. In ATR, the reputation of healers often spreads through word-of-mouth testimonies about their successes, a concept mirrored in NRMs' use of miracle stories to attract new followers. The principle of advertising healing and miracle stories demonstrates an aspect of global

phenomena being reinterpreted and adapted to the local context, leading to unique localised expressions. There was evidence of global and local influences between NRMs and ATR, where congregants shaped their religious practices. At the same time, the prophets played a role in shaping these NRMs' fundamental ethos and addressing the believers' spiritual and material needs.

The promised miracles also include financial breakthroughs in the form of miracle money[1], financial breakthroughs, gaining employment, and success at work. Hence, some people joined NRMs with the promise of economic transformation, which for many does happen. Such a promise made sense to a dejected community in Zimbabwe. More so, the emphasis on deliverance from economic hardships and the promise of financial breakthroughs in NRMs parallels ATR's practical approach to spirituality. ATR often addresses everyday issues, including economic and social challenges, through spiritual means. NRMs continue this tradition by promising and celebrating economic transformations and improved well-being. Within the alleged miracles, cultural hybridity was noted within the emphasis on spiritual healing and miracles, where NRMs often incorporate beliefs and practices that resonate with traditional African views on spirituality and healing. In addition, blending traditional African concepts of healing and miracles with Christian beliefs is both a convergence of cultural ideas and a syncretic practice.

*7.4. Use of Media*

One of the unique features of NRMs is the savvy use of new and old media to attract new members and advertise their ministries to broader society. Ahn (2013, pp. 67–72) argued that media could transform the individual through salvation and personal growth. The study noted that media productions, such as radio and television, featured prominently. Cultural hybridity was noted in the use of modern media technologies to spread the gospel and attract new members, showcasing an integration of contemporary global practices with local church activities, while the convergence of modern technology with traditional religious messaging was witnessed. PHD Ministries, like other prominent churches, has opened a television channel and has a fully backed production team that creates content for the church. Mashau and Kgatle (2019, p. 3) argue that television plays a fundamental role in that believers are encouraged to attend services through watching television, to the extent that others are encouraged to receive their healing through touching the television screen and connecting to the man of God. Touching of the television screen represents a cultural hybrid blend and glocalisation of traditional belief in the power of spiritual connection with the use of modern technology. This practice mirrors ATR's emphasis on tangible interactions with the spiritual realm, now extended through technological means. Interestingly, NRMs' use of television, radio, and social media platforms for outreach and community engagement reflects a modern adaptation of ATR's traditional methods of community gathering and oral storytelling. While ATR relies on direct, interpersonal communication within the community, NRMs extend this concept through modern media, reaching a broader audience while maintaining the communal essence of message dissemination.

The content is also shared using social media platforms such as Facebook, Twitter, Instagram, and WhatsApp as a means to attract new members. This fusion signifies how traditional religious concepts and beliefs are adapted to modern media formats, making them more accessible and relatable to contemporary audiences. Jiri, one of the participants discussing how she joined the PHD Ministries, shared:

I initially heard about the PHD Ministries and Prophet Walter Magaya's miracles through a friend in the township. Later on, I read a lot of comments about him on Facebook. I was impressed by the work the PHD Ministries was involved in. Soon after I joined, I was linked to the housing project I am currently contributing to, hoping I could one day own a house.

The platforms have provided another revenue stream for the ministries whilst providing jobs for young people who produce the content and run the media platforms of these ministries (Gifford 2004, p. 170). This approach to revenue generation and employment

opportunities reflects a pragmatic approach to religion, which aligns with ATR's focus on community well-being and practical solutions to societal issues, albeit adapted to a modern economic context.

Mapuranga (2013, p. 132), writing about religion, politics, and gender in Zimbabwe, argued that many people have also been influenced by the NRMs' use of bracelets and T-shirts with distinctive messages, photographs of leaders laminated on cars, and various branding methods, including television channels dedicated to these movements. The branded merchandise in NRMs above is reminiscent of ATR's use of symbols and artifacts in religious practices. However, NRMs have adapted this to modern branding techniques, creating visible symbols of faith and belonging that resonate in a contemporary societal context. Community members who joined such crusades display influential pictures and messages of their leaders sharing their experiences. The NRMs have created a media ecosystem that generates revenue from selling branded products. It is, however, evident that the use of media by NRMs showcases a syncretic blend and convergence with ATR, adapting traditional spiritual messages, community practices, and healing beliefs to modern communication technologies and methods. This adaptation ensures the relevance of these movements in the modern world and extends their reach and impact, blending the old with the new in innovative ways. This showcases the effects of globalisation and technological advancements, which have accelerated the process of cultural exchange and interactions, as noted by Cleveland et al. (2022).

*7.5. Announcements and Testimonies*

The overall observations showed that the time for announcements was at the end of each service. Testimonies of those healed or prospered in various ventures were provided. At the end of the testimonies, many congregants encouraged other congregants to remain connected to the prophet. One of the people giving testimonies told the congregants: 'Keep connected to the prophet, seed more, use anointing oil and holy water from the prophet.' The integration of announcements and testimonies in NRMs reflects syncretic elements with ATR, particularly in the use of storytelling, communal reinforcement of beliefs, and the emphasis on material prosperity as a sign of spiritual favour. Aderibigbe (2022) and Mahuika (2019) think ATR heavily relies on oral tradition and storytelling to transmit knowledge, values, and experiences. The use of personal testimonies, especially those relating to healing and prosperity, reflected a blend of African oral storytelling traditions with Christian testimonial practices. In addition, announcements and testimonies practice converge personal and cultural narratives with the broader narrative of the Christian faith.

In NRMs, oral tradition is continued by sharing testimonies during services. Similar to the oral narratives in ATR, these testimonies communicate personal experiences of healing and prosperity, reinforcing the power and efficacy of the religious practices and beliefs within the community. More so, the encouragement to stay connected to the prophet noted above, and the use of religious objects such as anointing oil and holy water parallels ATR's emphasis on the communal aspect of spirituality and the power of spiritual leaders and objects. Consequently, in ATR, community leaders, shamans, and healers play a pivotal role in guiding spiritual practices (Hitchcock 2023), a role that is mirrored in NRMs by the figure of the prophet.

Testimonies can be viewed as carefully choreographed stories encouraging other members to give more money to the ministry in the form of tithes and gifts. Focusing on prosperity and using testimonies to motivate congregants to contribute financially reflects both a syncretic blend of traditional beliefs and modern prosperity theology and merging of cultural practices to create a new form of expression. Adamo (2021) argued that material well-being is often seen as a sign of favour from the ancestors or deities in ATR. NRMs adopt this belief, portraying financial prosperity as a sign of divine favour and encouraging financial contributions to achieve it. It is also used to encourage new members to join the NRMs. Reverend Maka argued:

> There is a psychological impact on the poor person listening to the testimonies, which motivate the people to stay within the ministry, buying recommended religious objects with the hope that tomorrow will be their day of prosperity.

Observations during the study showed that prophets used the testimonies as a platform for alleged breakthroughs and the concept of seeding. The purpose of the testimonies was to show congregants that if one remains faithful to the prophet, one will be prosperous. The study observed a sense of the placebo effect[2] and the psychology of terrorism, which justifies and mandates certain behaviours, as noted by Borum (2011, p. 24). Hence, the use of testimonials to create a psychological impact and the observed placebo effect among congregants are modern interpretations of traditional beliefs in the power of faith and spirituality. In ATR, belief and faith are central to the effectiveness of spiritual practices, a concept adapted in NRMs to emphasise religious belief's psychological and emotional aspects. The element of global phenomena being reinterpreted and adapted in local context, leading to unique localised expressions as noted by Robbins (2017) was witnessed within the use of testimonies. The blend of NRMs and ATR also demonstrates the evolving nature of religious practices and beliefs, adapting traditional elements to contemporary religious contexts.

*7.6. Churchpreneurs*

The concept of 'churchpreneurship' in New Religious Movements (NRMs), as observed in Zimbabwe and other regions, exhibits convergence, glocalisation, cultural hybridity, and a specific type of cultural blending elements with African Traditional Religion (ATR), particularly in the commodification of spiritual practices and the focus on material prosperity. Observations between December 2019 and December 2020 showed that some pastors had been using their ministries to extract revenue from members through making merchandise of the gift of God, while some believers gave up the little they had with the hope that they would get it back a hundredfold. These leaders are now *churchpreneurs* who plant churches not because they want to save souls but to make money out of those who are vulnerable (Soboyejo 2016, p. 7). In Zimbabwe, with increased poverty and suffering, various entrepreneurial individuals have innovated biblical movements based on a specific gospel of prosperity as a means to amass wealth and social status (Mangena and Mhizha 2014). Churchpreneurship in the NRMs can influence the theological interpretation of the gospel of prosperity and blend religious practices with local cultural norms and elements renegotiated to meet the demands of prophets and pastors.

The study observed that after healing and deliverance, the PHD Ministries and GOM pastors would say, 'Give thanks to the Lord who has healed you. Give some money to the Lord; thank him for what he did for you.' This assertion suggests that *churchpreneurship* is characterised by sowing and seeding to the man of God. The emphasis on the 'gospel of prosperity' within NRMs aligns with certain aspects of ATR that relate to material wealth and success as signs of spiritual favour or ancestral blessings. However, in NRMs, this aspect is heavily commercialised, transforming spiritual blessings into a transactional relationship between the congregant and the spiritual leader, thereby demonstrating cultural elements being exchanged, transformed, and renegotiated, as noted in cultural hybridity and the reinterpretation and adaptation of the global phenomena into a unique localised expression.

Glocalisation was also noted in Jonathan Mbiriyamveka's article in *The Herald* of 27 July 2013, introducing the term 'gospreneurship' in Zimbabwe. He referred to the works of the NRMs as lucrative financial schemes. Thus, according to that article, 'gospreneurship' can mean the setting of the gospel mission as a platform for profiteering as in a business venture, which can be regarded as a 'latter-day, money-spinning family enterprise' (Marongwe and Maposa 2015, pp. 1–22). It was noted that some pastors became rich at some believers' expense. One of the participants argued: 'I am a typical example of those people who found themselves poorer after joining the PHD Ministries.' However, the PHD Ministries pastor and many other pastors have become rich while there is so much poverty in Zimbabwe. The

interplay of global and local influences in which local communities shape their religious practices benefited the prophets who play a role in shaping the fundamental ethos of NRMs.

Pastors have become rich by selling products such as anointing oil and holy water to believers desperate for a breakthrough. In South Africa, testimonies play a role within NRMs to the extent that the gospel of prosperity has become highly commercialised with the selling of handkerchiefs that make people successful, armbands, stickers for cars, anointing oil, prayer books, and holy water (Mashau and Kgatle 2019, p. 3). The selling of anointing oil, holy water, and other spiritual items by pastors in NRMs is akin to the traditional practices in ATR where spiritual healers might offer amulets, herbs, or talismans for protection, healing, or good fortune. However, in the context of NRMs, this practice has been adapted to a more commercialised form, reflecting a cultural hybridity of contemporary capitalist culture with religious practice. More so, convergence and syncretic blend of religious and economic systems, traditional spirituality, and modern capitalist practices within NRMs have replaced spiritual growth and moral sanctity, where preachers charge people for the cure of their ailments.

### 7.7. Sowing the Seed Preaching

Cultural hybridity within Sowing the Seed combines traditional African notions of reciprocal giving with Christian teachings on tithing and giving while converging the cultural practices of reciprocity with Christian doctrines of stewardship and prosperity. The concept of 'sowing the seed' preaching in New Religious Movements (NRMs) like the PHD Ministries and GOM, which emphasises giving to the church in expectation of abundant returns, presents a blend of traditional African religious beliefs and contemporary Christian prosperity theology. During the eight observation campaigns on the PHD Ministries and GOM and ten online sermons of the PHD Ministries on Yadah TV, it was noted that the pastors' preaching focused much on 'sowing the seed', a concept about giving to God and receiving back abundantly. When pastors preached, they would have a theme for the day. These themes included breaking the chains of the spirit of poverty, joy and prosperity, breakthrough, healing and deliverance, marriage, restoration, grace, competition, and various themes which were all linked to prosperity. The pastors made the following declarations such as 'I empower you through deliverance. Poverty is in mind, and I want you to move out of it' (Magaya, Sermon, 20 December 2017); 'Be Fruitful and Multiply…A right seed will feed a generation, change people, and create security for the unborn' (Magaya, Sermon, 31 December 2017); 'Being rich is not a sin. I can say I am a wealthy pastor in the spirit, and I want you to be rich like me' (Magaya, Sermon, 16 December 2018); 'Imagination is the more significant part of a miracle. Dream, imagine, and you can receive it. Hunt to give and never work for money but let money work for you. When you wake up every day, have the desire to succeed, rub off your past and you will see a change in your life' (Magaya, Sermon, 19 December 2018).

Other declarations were as follows: 'Compete with yourself every day. If you exercised 20 min yesterday, add 10 min today. If you were reading five books a month, add five more' (Magaya, Sermon, 19 December 2018); 'There is a war you must win for your success. I am seeing a great transition to prosperity' (Water Magaya Sermon accessed on 21 September 2018, https://youtu.be/LmWdWjBIvTE). 'When you give, you do not only give or receive money but what is behind the money. Pray when you give and pray when you receive' (Water Magaya Sermon accessed on 27 July 2018, https://youtu.be/7VSzJzgFBOo). 'When you are under attack by the spirit of poverty, it attacks what is in your pocket so that you end up saying I almost made it' (Water Magaya Sermon accessed on 16 October 2019 on https://youtu.be/lbaeZgnKUnE) 'There is joy and prosperity in the name of Jesus' (Manenji, Sermon, 21 December 2020); 'Meditation helps you to attract what you want, and you shall get it. Meditation is the key to prosperity' (Manenji, Sermon, 21 December 2021); 'When you want a breakthrough for money, healing, and a house, you need to have verses to memorise' (Manenji, Sermon, 23 December 2020); 'Two things show that there is love. It is giving and forgiving. God forgave and gave us a sign of love. When you love, then

be prepared to give something back. Give something back to God' (Manenji, Sermon, 23 December 2021).

The above quotations suggest that the PHD Ministries and GOM emphasised giving to the pastor so that God would abundantly reward or give back to believers. It is believed that God will multiply the money given and return it to the giver. The focus on financial giving as a spiritual act that leads to material rewards reflects a hybrid influence on theological interpretation of the gospel of prosperity and syncretic adaptation of ATR in particular or even other religious traditions where offerings and sacrifices are made to deities or ancestors in the hope of receiving blessings. However, in NRMs, this practice is heavily influenced by the prosperity gospel, which directly links financial contributions to divine favour and material wealth. Teaching on sowing the seed derives from 2 Corinthians 9:6–11, which teaches that 'whoever sows sparingly will also reap sparingly, and whoever sows generously will reap generously.' In their preaching, the PHD Ministries and GOM encouraged believers to sow or give generously to the Church to receive back abundantly. Using biblical passages such as 2 Corinthians 9:6–11 to reinforce the concept of giving and receiving aligns with Christian doctrines and other religious and non-religious traditions. However, it is interpreted in a way that mirrors many religious traditions and traditional African concepts of reciprocity and balance in the spiritual realm. In addition, a combination of different cultural and religious elements has been witnessed to form a cohesive whole (Pieterse 2019) which created a new form of religious expression of sowing the seed.

Prophet Walter Magaya emphasised that he was rich, and his believers would be rich like him if they also gave to the Church. This type of preaching confirms Coleman's (2011, pp. 23–45) and Togarasei's (2011, p. 341) writing about the prosperity gospel in the African context, arguing that pastors always urge followers to give to the Church or sow the seed generously, trusting that God will abundantly reward them for an act of faith. Such preaching in NRMs often involves empowering believers through faith, visualisation, and positive affirmations. This approach converges traditional African emphasis on the power of words and thoughts with modern Christian teachings on faith and prosperity.

## 8. Correlations within the Characteristic Features of NRMs

The above section showed that there needs to be more clarity around the scholarly debates on cultural hybridity, convergence, glocalisation, and syncretism, making it difficult to describe the religious and cultural practices constituting these concepts. However, the characteristic features of NRMs intertwine to form a unique religious expression that reflects cultural hybridity, convergence, glocalisation, and syncretism. NRMs often exhibit distinctive characteristics that set them apart from traditional religious practices. These characteristics, such as unique worship styles, a focus on alleged miracles, strategic use of media, reliance on personal announcements and testimonies, the rise of 'churchpreneurs', and the practice of 'sowing the seed', are not only interconnected but also crucial in defining the identity and appeal of these movements. The worship style in NRMs often breaks away from conventional liturgies, incorporating contemporary music, technology, and often emotive and charismatic expressions. This modernised and often experiential approach to worship is designed to resonate with a younger, more contemporary audience, fostering a sense of community and belonging that is both refreshing and relatable. The emphasis on alleged miracles, particularly those related to healing and financial prosperity, taps into followers' existential needs and hopes, often as a powerful tool for conversion and retention. More so, the savvy use of media, including social media platforms, television, and other digital channels, is another defining trait of NRMs. This helps disseminate their message to a broader audience and creates a brand-like appeal, which is essential in a media-saturated world. Similarly, personal announcements and testimonies, often shared during services or through media channels, authenticate the movement's claims, providing personal and emotive narratives that make sense to its followers.

The rise of 'churchpreneurs'—leaders who blend spiritual guidance with business acumen—is a distinct phenomenon within NRMs. These leaders often promote a prosperity

gospel, which blends religious doctrine with principles of wealth and success. This is particularly appealing to followers in economically challenging environments. This ties in with the concept of 'sowing the seed', in which followers are encouraged to make financial contributions with the expectation of divine financial returns. This practice not only fuels the economic engine of the movement but also reinforces the prosperity narrative. In short, these characteristics are not isolated elements but form a synergistic web that defines the operational and theological framework of New Religious Movements. They consciously adapt to contemporary culture and social dynamics to address their followers' spiritual, emotional, and material aspirations in an increasingly complex and globalised world. What is clear is that these elements demonstrate a dynamic blend of traditional beliefs and modern cultural influences, highlighting the NRMs' adaptive and syncretic nature in a globalised context.

## 9. Diverging Syncretic Elements between NRMs' Features and ATR

Syncretism in religious practices often involves blending and merging different traditions to form a new, hybrid religious identity. This syncretism reveals converging and diverging elements in the context of NRMs and ATR in Zimbabwe. A critical analysis of these divergences is essential to understanding NRMs' unique characteristics and impacts in Harare, Zimbabwe. Regarding material prosperity versus communal well-being, it was noted that ATR traditionally emphasises communal well-being and spiritual harmony within the community. The NRMs, however, heavily focus on individual material prosperity, particularly in the context of the prosperity gospel. This shift from community-centric to individual-centric prosperity marks a significant divergence, often leading to social stratification within the religious community. In addition, it was noted that in ATR, offerings and sacrifices are made to deities or ancestors, often as a form of respect, gratitude, or seeking guidance, without a direct expectation of material return. Conversely, NRMs adopt a more transactional approach to spirituality, where giving is often linked to expected material rewards. This transactional nature represents a shift from the traditional values of ATR, which focus more on spiritual balance and ethical conduct.

Observations from research showed that the seating arrangement at the PHD Ministries is hierarchical and is based on who has paid more to the ministry. For example, those who paid USD 350 to stay in the hotel belonging to Prophet Walter Magaya would sit close to him during services, followed by those who paid USD 100. Those who did not pay anything always sat far away from the pastor. One of the participants, Gari, argued: 'The seating arrangement within the PHD Ministries is simple. There is a place for the rich, not so rich and the ordinary believers who are the biggest population.' This seating arrangement, however, seems to suggest that congregants are not equal; some are more important than others. GOM, however, did not have a special seating arrangement, as all congregants were treated as equals, although the seating arrangement could change as the ministry grows. The seating arrangement at the PHD Ministries, suggested that the gospel of prosperity is instrumental to managing class relations. This confirms the writings of Wrenn (2019, p. 427) about the prosperity gospel and neoliberalism, arguing that capitalism holds precise class lines: for the upper class, it further justifies their place in the hierarchy; for the middle class, it affirms their aspirations and opens the perception of possibilities; and for the poor, the prosperity gospel gives hope.

The seating arrangement at the PHD Ministries showed a Janus face[3] in which secure networks and reciprocities were built amongst the rich and middle class. In contrast, the same networks and reciprocities between the rich and the poor, who could not associate and build social cohesion, were destroyed or eliminated. The seating arrangement caused exclusion and hindered economic progress among the believers. In this way, the rich became social capitalists, creating networks that excluded the ministry's poorer members. More so, the hierarchical seating arrangement observed in the PHD Ministries, where financial contributions determine proximity to the spiritual leader, presents a syncretic blend between contemporary capitalist structures and traditional African religious practices,

albeit in a way that diverges from the communal and egalitarian ethos typically found in African Traditional Religion (ATR). In other words, it was also noted that traditional ATR practices typically emphasise egalitarianism within spiritual gatherings. However, some NRMs' hierarchical seating arrangements and class-based treatment reflect a departure from these egalitarian principles. Based on financial contribution, this hierarchy contrasts with ATR's more inclusive and community-oriented approach.

Commercialising spiritual items such as anointing oil, holy water, and other religious merchandise in NRMs significantly diverges from ATR practices. While ATR involves the use of spiritual objects, the commodification and sale of these items in a capitalist manner is not traditionally aligned with ATR principles. In ATR, spiritual leaders or shamans are traditionally seen as guides and healers, who are deeply integrated into the community's life. In NRMs, the role of spiritual leaders can sometimes shift towards a more authoritarian or celebrity-like status, with a strong emphasis on their wealth and personal success, diverging from the traditional role envisioned in ATR.

NRMs targeting vulnerable individuals for financial gain diverges from ATR's communal and ethical principles. While traditional African spirituality emphasises community support and ethical integrity, some NRMs exploit these principles for personal gain, revealing a syncretic yet problematic aspect of these movements. The shift towards capitalistic approaches within NRMs, where spiritual growth and moral sanctity are overshadowed by financial transactions for spiritual services, indicates a syncretic blending of religious and economic systems. This blending, however, often strays from ATR's focus on spiritual integrity and communal well-being. In other words, the emergence of 'churchpreneurship' in NRMs showcases a syncretic mix of traditional African spiritual elements with modern capitalist practices. While this blend reflects the adaptive nature of religious practices, it raises ethical concerns regarding the commercialisation of spirituality and the potential exploitation of believers' faith and vulnerabilities.

## 10. Contribution of NRMs to Social Cohesion and Information Sharing

The third argument within this article postulates that the cultural hybridity, convergence, glocalisation, and syncretism in the NRMs provide pathways to invaluable networks. Despite the above divergences, NRMs in Harare have contributed to social cohesion and information sharing, positively impacting spiritual and material prosperity and holistic community development. The contribution of NRMs to social cohesion and information sharing in Harare, Zimbabwe, is multi-faceted. By addressing both spiritual and material needs, these movements have fostered new forms of social networks and communities, playing a crucial role in the holistic development of individuals and communities within a rapidly changing society.

### 10.1. History of Pentecostalism in Zimbabwe and Networks Created

Social cohesion and information sharing were witnessed even within the history and evolution of Pentecostalism in Zimbabwe, particularly its response to poverty and unemployment. The history of Pentecostalism offered a compelling insight into how religious movements can play a pivotal role in community formation and empowerment. For example, due to mining activities in South Africa, many migrants from Zimbabwe, Malawi, and Zambia were hosted and were believed to have embraced Pentecostalism in South Africa. Upon returning home, it was believed that the migrants preached the same gospel in Zimbabwe (Maxwell 1998; Togarasei 2005; Sundkler and Steed 2000). However, the use of modern media within the NRMs facilitated widespread information and resource sharing. The strategic use of NRMs' social media, television, and radio broadcasts revolutionised information sharing in Harare. These platforms make it easier to spread spiritual teachings, health information, and economic opportunities. This contributes significantly to the education and empowerment of individuals and communities, particularly in areas in which access to information might otherwise be limited.

As part of a global phenomenon, Pentecostalism in Zimbabwe has distinctive African characteristics, blending spiritual practices with local cultures and societal needs. More so, Pentecostalism's growth in Zimbabwe, notably through the Apostolic Faith Mission (AFM), Assemblies of God (AOG), Family of God (FOG), and Zimbabwe Assemblies of God Africa (ZAOGA), highlights its adaptive and responsive nature to local contexts (Matikiti 2017, pp. 138–42; Gifford 1988, p. 2; Mumford 2012, p. 372; Bishau 2015, p. 5). These Pentecostal churches have been instrumental in addressing the spiritual, social, and economic challenges faced by their congregations, particularly in the face of poverty and unemployment. NRMs have been influential in creating new social networks and communities. At the same time, Maxwell (2006, p. 38) and Togarasei (2005, p. 2; 2016, p. 5) argue that Pentecostalism in Zimbabwe gave birth to several African Initiated or Instituted Churches (AICs) and the rise of other Pentecostal churches. Even though there has been an overlap, AICs which evolved before, after, and within classical Pentecostalism flourished within their focus on communality and the amalgamation of facets of Christianity with the culture of the local people (Hastings 1994, p. 118; Sundkler and Steed 2000, p. 816). More so, Pentecostalism's contribution to community formation was witnessed from the official recognition of AFM in 1943, leading to the proper coordination and administration of the church, activities, establishment of order, and discipline, which also led some members to leave the church voluntarily or by expulsion (Togarasei 2016, p. 5). The mass exodus of people from the AFM saw other people form apostolic churches, which became the AICs, developing theologies that differed from the AFM teaching.

Critical to Pentecostalism in Zimbabwe is its focus on communal empowerment and personal transformation, similar to how NRMs have established themselves as significant social institutions in Harare. Organising regular gatherings, social events, and group activities allows individuals to interact, share experiences, and support each other. This interaction fosters a sense of belonging and community. The Pentecostal churches emphasise the workings of the Holy Spirit, speaking in tongues, miracles, healing, and prophecy (Anderson 2004, pp. 103–4; Togarasei 2005, p. 349; Chibango 2016, p. 71). This spiritual emphasis offers a sense of hope and empowerment, which is essential in communities grappling with economic hardships. However, in societies where traditional social structures may weaken, NRMs fill an essential gap, creating new forms of social capital and reinforcing communal bonds. In addition, NRMs provide spiritual and psychological support to their members, offering a sense of purpose, direction, and hope. Such support is invaluable in the context of economic hardship or social instability. It contributes to individual resilience and community solidarity, helping people navigate challenging times with a strengthened sense of self and community.

Pentecostal churches have also been active in community development and social services. For example, ZAOGA Forward in Faith has extended its reach beyond spiritual guidance, running schools, orphanages, hospitals, and even a university (Musoni 2013, p. 80). Gukurume (2021, p. 34) argues NRMs have influenced assisting families in meeting their basic needs, income-generating projects, educational support such as scholarship, and community infrastructural development. These initiatives demonstrate the NRMs' commitment to practical solutions to societal challenges. The holistic approach to community service demonstrates the movement's commitment to addressing not just the spiritual but also the material needs of the people. By addressing the people's material needs, NRMs' response is uniquely characterised by promoting entrepreneurship and self-reliance among its followers. The 'gospel of prosperity' preached within neo-Pentecostal churches or some of the NRMs motivates individuals to pursue economic success, often creating businesses and job opportunities. This approach provides a pathway out of poverty and instils a sense of agency and empowerment among the religion's followers. The inclusivity and appeal of Pentecostalism to a broad demographic, including the youth, play a significant role in community formation. By attracting young people and nurturing their aspirations for success, the NRMs are shaping a new generation of empowered individuals who are spiritually and economically equipped to contribute to their communities.

### 10.2. The Characteristic Features of NRMs and Networks Created

The characteristic features of New Religious Movements (NRMs), such as diverse hymns, dynamic praying styles, and the emphasis on miracles and media use, serve as prime examples of cultural hybridity, convergence, glocalisation, and syncretism within religious practices. These NRMs adeptly blend traditional religious elements with modern cultural influences, creating a unique religious expression that resonates with contemporary audiences. This synthesis includes incorporating charismatic worship styles and breaking traditional liturgies by integrating contemporary music and technology. Such an approach appeals to younger demographics and fosters a sense of community belonging. Within the selected NRMs, internal groups, such as choir and musical groups, created spaces for socialisation and social cohesion amongst the believers, where they started exchanging contacts and sharing ideas and business deals. In a focus-group discussion (FGD 1), Tim from GOM pointed out that 'through choir, I managed to create friends, and we have started small projects such as buying and selling non-perishables such as roasted corn and water around the ministry's property.' Another participant within the PHD Ministries, Jim, reported, 'Some of our youth who are within the choir are linked with professional gospel singers, and they sometimes perform as curtain-raisers at the musical shows.' Hence, social cohesion was created by youth who could network, collaborate, and interact with one another before, during, and after choir practice and ministries' services.

In the realm of NRMs, the occurrence of alleged miracles, particularly those related to healing and financial prosperity, addresses followers' existential needs and hopes. This aspect of NRMs is often a powerful tool for conversion and retention, tapping into deep-seated human desires and aspirations. The strategic use of media, notably social media platforms and digital channels, further exemplifies the glocalisation and convergence within these movements. This modern communication approach allows NRMs to extend their reach beyond traditional boundaries, making their message more accessible and appealing to a broader, global audience. The emergence of 'churchpreneurs' within NRMs, who combine spiritual guidance with business acumen, underscores the syncretic nature of these movements. They often promote a prosperity gospel, combining religious teachings with wealth and success principles. This is particularly appealing in economically challenging environments and aligns with 'sowing the seed', encouraging financial contributions with the promise of divine financial returns. Such practices not only support the economic sustainability of the movements but also reinforce the narrative of prosperity, which is a vital part of their appeal.

Interestingly, PHD Ministries and GOM encouraged entrepreneurial activities and economic empowerment among their members. By promoting ideas such as 'sowing the seed' and prosperity theology, these movements have motivated individuals to pursue economic goals, leading to material prosperity for some of the believers. The hierarchical structures of PHD Ministries and GOM, while reflecting a divergence from ATR's egalitarianism, also facilitated the formation of social capital among wealthier members. This networking led to business opportunities and economic collaborations, benefiting the involved individuals and, potentially, their wider communities. More so, the engagement of selected NRMs in community development projects and charity work exemplifies their contribution to the holistic development of communities. Consequently, NRMs often function as platforms for cross-cultural and interfaith dialogue, fostering understanding and tolerance among different groups. This aspect contributes to social cohesion by promoting respect for diversity and encouraging peaceful coexistence.

The syncretic and hybrid practices within NRMs create gateways to invaluable networks, fostering social cohesion and the sharing of critical information. Through these networks, NRMs facilitate the holistic development of individuals and communities, highlighting the intricate interplay between spirituality and material prosperity. Through their emphasis on spiritual empowerment, community service, entrepreneurial encouragement, and philanthropic activities, NRMs have played a significant role in community formation and empowerment. They offer a dynamic model of how religious movements can positively

impact both the spiritual and material aspects of life, contributing to communities' overall development and resilience. These initiatives address both spiritual needs and material necessities, thereby playing a crucial role in enhancing communities' overall well-being and cohesion. It was noted that the selected NRMs engaged in community development projects and charity work. These initiatives contribute to the holistic development of communities, addressing both spiritual needs and material necessities. In other words, NRMs in Harare have played a significant role in promoting social cohesion and information sharing. Their activities have not only catered to spiritual and psychological needs but have also facilitated economic empowerment, community development, and the fostering of new social networks. These contributions are especially vital in a rapidly changing society, in which traditional structures and norms are continually redefined.

## 11. Conclusions

In conclusion, this article has provided an insightful analysis of the interplay between cultural hybridity, convergence, glocalisation, and syncretism in New Religious Movements (NRMs) and their profound impact on both spiritual and material life in Harare, Zimbabwe. The exploration of NRMs, especially in their alignment with African Traditional Religion (ATR) tenets, reveals a rich tapestry of religious expression that is unique and complex. Prophets and pastors within these movements have been instrumental in shaping not just the spiritual ethos but also influencing the socio-economic practices of their followers. This influence extends to practices such as exorcism, worship, healing, and deliverance, underscoring a deep-rooted syncretism that blends traditional African spirituality with modern Christian beliefs. This fusion, however, presents both challenges and opportunities. On the one hand, it raises questions about the authenticity of religious experiences and the potential for exploiting believers' faith, particularly in the context of prosperity theology. On the other hand, the positive aspects of these NRMs cannot be overlooked. They have played a significant role in fostering social cohesion and information sharing, contributing immensely to the holistic development of individuals and communities in Harare. This contribution is particularly evident in how these movements have become integral to spiritual and material prosperity in a vibrant and dynamic religious landscape.

The findings of this article highlight the richness of NRMs in terms of their ability to adapt and resonate within diverse environments. They have redefined the religious and socio-economic fabric of NRMs in Harare, offering a window into the complex interplay between spirituality, material prosperity, and religious traditions. This study opens new avenues for future research in religious studies, inviting a deeper exploration into the implications of religious practices in contemporary societies. In essence, this article serves as a critical reference point for understanding NRMs' multifaceted roles and impacts. It emphasises their capacity to root themselves and translate their ideas across various environments, reflecting the broader context of global religious practices. This investigation into NRMs enriches our understanding of religious dynamics in Harare, Zimbabwe, and contributes to the broader discourse on religion's role in modern society.

**Funding:** This research received no external funding.

**Institutional Review Board Statement:** The protocol has been granted EXEMPTION FROM ETHICS REVIEW (UKZN Protocol reference number: HSS/0514/017D) because the study did not involve humans or animals.

**Informed Consent Statement:** Not applicable.

**Data Availability Statement:** Data sources were various articles in the reference list and due to ethical restrictions personal data is not available due to privacy.

**Conflicts of Interest:** The author declares no conflict of interest.

## Notes

1     https://www.africaontheblog.org/miracle-money-gold-and-abortions-zimbabwes-wacky-prophets/ (accessed on: 12 October 2023).
2     This means that what heals some people is not the medicine administered to them, but the faith in the person who administered the medicine.
3     Janus was a Roman deity depicted with two faces, one looking forward and one looking backwards. The term Janus face, therefore, refers to two contrasting or different aspects or characteristics.

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
