# Peer review of "Exploring Cultural Hybridity Branded by Convergence and Syncretism in the Characteristic Features of the Pentecostal Charismatic Churches in Zimbabwe: Implications for Spiritual and Material Well-Being"

_religions, doi:10.3390/rel15010102_

Round 1
Reviewer 1 Report
Comments and Suggestions for Authors
The article joins and contributes to our growing knowledge of the varieties of charismatic Christianity throughout Africa. It clearly highlights points of contact between African traditional religions and neo-Pentecostal theologies and practices, particularly the prosperity gospel. I agree with the author’s conclusion that the richness of Pentecostalism rests in its ability to take root and translate its ideas in diverse environments. This has been a hallmark of the movement from the very beginning and, as many scholars note, key to its worldwide success. The article’s main contribution to extant literature on the topic is the detailed focus on specific churches in the context of Zimbabwe. However, there are several important issues that must be addressed prior to publication.
The first, and most pressing, is the need to present a clear discussion of its rationale for employing such a dated reference to ‘syncretism’ as opposed to more recent adoptions of ‘hybridity.’ There is significant scholarly debate around ‘syncretism,’ ‘hybridity’, and ‘convergence’ that is not addressed in the paper. This is not to suggest that the author’s use of ‘syncretism’ is incorrect. Sharpe’s work was crucial nearly half a century ago, but no longer reflects the current state of the field. In short, the article would benefit from the inclusion of a discussion of the wider trends in scholarship around these terms if only to clarify why the author chose ‘syncretism’ as the most descriptive lens through which to approach the processes identified in the paper. The author might consider reviewing Ackermann’s research (2011) on cultural hybridity or even more recent scholarship on cultural convergence.
In line with my previous suggestion, is a concern about the nature of the examples provided to support the author’s claim of syncretism. Much of the work focuses on specific practices (hymns, singing, raising hands, etc.) which do not reflect syncretic formations – even according to the way the author defines the term (“where distinct religious elements merge harmoniously” p. 1, line 43-4). These types of charismatic practices are not unique to the Pentecostal churches reviewed in the article, or even to Pentecostal churches in Africa. The practice of giving back to the church through financial contributions, for example, which the author refers to as akin to the ‘reciprocity’ exercised by ATRs, is characteristic of many religious traditions, not just ATRs or Christianity. Most of the practices identified in the groups under study here are defining traits of Pentecostalism more broadly. The reader is left wondering – Where exactly are the elements of the ATRs in these churches that would make them syncretic representations? The practices the author describes in the article seem to be more in line with analyses that focus on ‘convergence’ between neo-Pentecostalism and local contexts and ways of knowing. For instance, the primacy of oral testimony in the worship styles of the Pentecostal churches, which the author suggests is evidence of syncretism between the ATRs and the churches, appears simply as a way the two worldviews align or ‘converge’ rather than the ways they have exerted influence on one another to produce a fusion. This issue might be addressed by including some analysis of the idea of ‘glocalization’ which looks deeper at the ways religious traditions are transformed in their local contexts. The work of Joel Robbins has been particularly useful for thinking about this process.
There is also some confusion regarding the author’s use of the term New Religious Movements (NRMs). It can be (and has been) argued that neo-Pentecostal churches are new religious movements. However, not all new religious movements are Pentecostal. The author conflates the two, using NRMs as a stand in for describing localized formations of Pentecostal practice. In fact, I do not believe the constant references to NRMs are completely necessary given the focus of the paper (syncretism).
The section on the contributions of NRMs to social cohesion is underdeveloped and requires further analysis. The author does not provide any scholarly references about Pentecostalism’s contribution to community formation and empowerment. The author might consider reviewing David Maxwell’s work on Pentecostalism, neo-liberalism, and Zimbabwe (2005) as a starting point.
Finally, the article needs minor stylistic and grammatical revision. Examples include (this list is not exhaustive):
p. 2 ln 77 – a period is missing after ‘victory’
p. 3 ln 130 – the capitalization of ‘Ancestor’
p. 12 ln 60 – “this confirms the writings of Wrenn’s about the prosperity gospel” should be revised
Comments on the Quality of English LanguageMinor grammatical and typographical revisions needed.
Author Response
"Please see the attachment"

Reviewer 2 Report
Comments and Suggestions for Authors
This is an interesting topic and relevant to the overall theme of Religions Journal. The author demonstrates passion for the topic and knowledge in the subject area. However, the work falls short in critical analysis of both the chosen topic and overall subject. There is need for discussion of the word "Syncretism" as understood and used in the manuscript. More is needed to conclude that this paper provides profound insights on the subject nor contributes substantially to the current comprehension as stated by author.
Most of the the characteristics observed in the NRM are not unique to Pentecostal Charismatic churches, but have been observed by authors of ATR as common to most Christian (and in some cases religious) faith traditions of African heritage. More research is needed to make the distinct correlation between those characteristics and the NRM.
If possible, identify the exact qualitative research method used.
Author Response
"Please see the attachment"

Round 2
Reviewer 1 Report
Comments and Suggestions for Authors
The author has addressed many of my concerns around the use of terms such as syncretism, cultural hybridity, and convergence. The author provided a thorough explanation at the beginning which I find ground their analysis later in the paper.
I still hold some reservations about whether the practices described by the author constitute syncretic or hybridized practices among Pentecostal-Charismatic churches. However, this may be a reflection of the ambiguity around the scholarly debates of the definitions of those terms.
I suggest further minor edits concerning the syntax and language, especially in new sections added by the author. Certain phrases (ie. 'scholarly debates', 'in the context of', etc.) are repetitive and distract from the analysis. Moreover, the author's use of terms like 'cultural hybridity elements' (again overused as a phrase, not a term) should be rewritten to something akin to 'elements of cultural hybridity'. Also, the bibliography lacks a reference to Ackermann that is cited in the paper.
Comments on the Quality of English LanguageMinor changes to syntax and diction are necessary before publication.
Author Response
Open Review 1 second round
Comments and Suggestions for Authors
Reviewer :The author has addressed many of my concerns around the use of terms such as syncretism, cultural hybridity, and convergence. The author provided a thorough explanation at the beginning which I find ground their analysis later in the paper.
Response: I have also addressed the following issues:
- The arguments and discussion of findings have been made coherent, balanced and compelling throughout the article. In the introduction, page 3, I presented three main arguments highlighted in blue, which guided the article's structure.
- I have supported my findings and conclusions with secondary literature references.
Reviewer’s comments: I still hold some reservations about whether the practices described by the author constitute syncretic or hybridized practices among Pentecostal-Charismatic churches. However, this may be a reflection of the ambiguity around the scholarly debates of the definitions of those terms.
Response: I have made it clear that the complexity and ambiguity around the debates of the definitions of these terms (cultural hybridity, convergence, glocalization and syncretism) also present challenges in understanding the religious practices and how they constitute syncretic or hybridised practices among Pentecostal-Charismatic Churches. See page 4, section 4.
Reviewer: I suggest further minor edits concerning the syntax and language, especially in new sections added by the author. Certain phrases (ie. 'scholarly debates', 'in the context of', etc.) are repetitive and distract from the analysis. Fixed Moreover, the author's use of terms like 'cultural hybridity elements' (again overused as a phrase, not a term) should be rewritten to something akin to 'elements of cultural hybridity'. Also, the bibliography lacks a reference to Ackermann that is cited in the paper.
Response: I have made extensive edits concerning grammar and language in the sections highlighted and the whole article. Repetitions on ‘in the context of’ and other repetitions or overuse of terms have been corrected throughout the article. Ackermann (2011) has also been added to the reference list.
Reviewer : Minor changes to syntax and diction are necessary before publication.
Response: I have made edits and changes throughout the article. All changes and corrections made are highlighted in blue.
Reviewer 2 Report
Comments and Suggestions for Authors
In the first review I should have clarified that you need to identify the approach of the qualitative research method used. Not important but if is a distinct approach, it may help mentioning.
Author Response
Comments and Suggestions for Authors
Response: I have clarified how the results will be presented in the introduction, drawing from the three main arguments highlighted on page 4. The conclusion has also been revised. All the changes within the article are highlighted in blue.
Reviewer: In the first review I should have clarified that you need to identify the approach of the qualitative research method used. Not important but if is a distinct approach, it may help mentioning.
Response: I have revised the part highlighting that the study used an ethnographic approach to collect data through interviews, Focus Group Discussions (FGDs) and observations within the qualitative methodology. See page 6, section 5.